# DNA-Templated Silver Nanoclusters as Dual-Mode Sensitive Probes for Self-Powered Biosensor Fueled by Glucose

**DOI:** 10.3390/nano13081299

**Published:** 2023-04-07

**Authors:** Akhilesh Kumar Gupta, Alexey V. Krasnoslobodtsev

**Affiliations:** Department of Physics, University of Nebraska at Omaha, Omaha, NE 68182, USA

**Keywords:** glucose biosensor, silver nanoclusters, fluorescence, self-powered biosensor, next-generation healthcare technology

## Abstract

Nanomaterials have been extensively explored in developing sensors due to their unique properties, contributing to the development of reliable sensor designs with improved sensitivity and specificity. Herein, we propose the construction of a fluorescent/electrochemical dual-mode self-powered biosensor for advanced biosensing using DNA-templated silver nanoclusters (AgNCs@DNA). AgNC@DNA, due to its small size, exhibits advantageous characteristics as an optical probe. We investigated the sensing efficacy of AgNCs@DNA as a fluorescent probe for glucose detection. Fluorescence emitted by AgNCs@DNA served as the readout signal as a response to more H_2_O_2_ being generated by glucose oxidase for increasing glucose levels. The second readout signal of this dual-mode biosensor was utilized via the electrochemical route, where AgNCs served as charge mediators between the glucose oxidase (GOx) enzyme and carbon working electrode during the oxidation process of glucose catalyzed by GOx. The developed biosensor features low-level limits of detection (LODs), ~23 μM for optical and ~29 μM for electrochemical readout, which are much lower than the typical glucose concentrations found in body fluids, including blood, urine, tears, and sweat. The low LODs, simultaneous utilization of different readout strategies, and self-powered design demonstrated in this study open new prospects for developing next-generation biosensor devices.

## 1. Introduction

The use of nanomaterials in the construction of biosensors leads to better designs and more capable sensing mechanisms. Therefore, nanomaterials have enabled the development of highly sensitive and selective biosensors with improved performance [1,2,3]. These biosensors are becoming more capable of detecting very low levels of analytes due to the enhanced surface-to-volume ratio of nanomaterials [4,5]. It is also possible to miniaturize biosensors due to the small size of nanomaterials, which contributes to the development of portable and point-of-care technologies [6]. Improved performance and miniaturization lead to reduced costs of sensor use, thus providing prospects for personalized medicine in that such sensors can be used in resource-limited environments for the routine monitoring of patients [5]. The next step in developing biosensors is to reduce their reliance on external power sources, making them as energy efficient as possible [7]. Ideally, biosensors need to be self-powered, harnessing available sources of energy such as mechanical (piezoelectric or triboelectric) [8], thermal (thermoelectric devices) [9], or chemical (biofuel cells) sources [7].

Biofuel cells (BFCs) are ecofriendly electrochemical systems that may generate electrical energy by employing organic materials produced during metabolic processes as fuel and are a renewable biological catalyst, usually a microorganism or an enzyme [10]. Enzymatic biofuel cells are probably the most promising BFCs, attracting attention due to their ability to generate electrical energy at low temperatures, their physiological pH levels, and their high biocompatibility. Enzymes are immobilized on the electrode’s surface, opening up opportunities for developing miniaturized systems for powering electronic devices as implantable and self-powered electrochemical biosensors [11,12]. The main advantage of BFCs is that they require no external power supply while using body metabolites as fuel, such as glucose, lactate, cholesterol, ethanol, and others [13,14,15,16,17]. Glucose is a very attractive fuel source since it is constantly supplied by metabolic activities in the human body in relatively high amounts [18]. Recently, research efforts have focused on developing membraneless BFCs that can deliver electrical energy using glucose oxidation at an anode and O_2_ or H_2_O_2_ reduction at a cathode [19,20].

The performance of BFCs has been significantly increased during the past decade by utilizing nanomaterials such as carbon nanotubes (CNTs) [21], graphene oxide (GO) [22], noble metal nanoparticles [23], and conjugated polymers (CPs) [24]. For example, the use of carbon nanotubes as a support material for enzymes has been shown to increase the stability and activity of the enzymes in fuel cells. The utilization of graphene has improved the electrode’s conductivity and increased the fuel cell’s efficiency. These materials are often biocompatible, electrically conductive, and have a wide surface area. Their use improves the efficiency of electron transfer and the amplitude of the generated electrical signal, and they frequently provide a stable matrix for immobilization enzymes. Nanomaterials with a large surface area can boost enzyme loading and improve the activity and stability of immobilized enzymes, enhancing the performance of BFCs [25].

Recently, DNA-templated AgNCs (AgNCs@DNA) have emerged as attractive nanomaterials of supra-atomic size with great potential for use in practical applications. Due to their unique properties such as superior biocompatibility, water solubility, high-fluorescence quantum yields, and tunable fluorescence emission, AgNCs@DNA have been utilized in biosensing and bioimaging [26,27,28]. In addition to the many reports on their optical properties, other properties of AgNCs@DNA associated with the unique structure of electronic states have been reported. For example, antibacterial activity has been recently reported, which was discussed in connection with a distinct color change, a unique charge state of AgNCs@DNA leading to the production of free radicals and possibly singlet oxygen [29,30,31]. Researchers have also used phenomena such as guanine-rich DNA sequence-activated fluorescence enhancement [28,32,33,34], aggregation-induced emission (AIE) [35], and photoinduced electron transfer (PET) [36] to design biosensing strategies using silver nanoclusters. The sensitivity of the fluorescence of AgNCs@DNA to environmental conditions prompted the development of sensors that are sensitive to heavy metals [27,37,38,39], miRNA [28,40,41,42], and even small molecules such as dopamine [43], melanin [44], and hydroquinone [45]. All previously reported uses of AgNCs@DNA have exploited their fluorescence properties to build optical sensors. Herein, we report on a novel use of AgNCs@DNA in a glucose detection strategy that combines two readout signals: optical (fluorescence) and electrochemical (current). This shall expand the use of these nanomaterials in next-generation biosensor devices for a wide range of applications.

## 2. Materials and Methods

### 2.1. Chemicals

DNA oligonucleotides were purchased from Integrated DNA Technologies (IDT Inc., Coralville, IA, USA) as desalted products and were used without further purification. Nuclease-free water was obtained from IDT (Coralville, IA, USA). Sodium borohydride was purchased from TCI America Inc. (Portland, OR, USA). GOx was obtained from Aspergillus niger, and N-(3-dimethylaminopropyl)-N′-ethylcarbodiimide hydrochloride (EDC), Pyrrole-2-carboxylic acid (PCA), and D-(+)-glucose were purchased from Sigma Aldrich. Hydrogen peroxide (H_2_O_2_) was obtained from Chempur, and N-hydroxysuccinimide (NHS) was obtained from Merck. All chemicals were of analytical grade. All aqueous solutions were prepared in ultrahigh-quality (UHQ) water. In addition, a glucose solution was prepared at least 24 h before use to allow the glucose to undergo mutarotation and to reach equilibrium between α and β forms. A 10.0 mg/mL solution of GOx was freshly prepared in a sodium acetate and phosphate buffer solution, and PCA was prepared in 200 proof ethanol.

### 2.2. Instrumentation

The excitation and emission spectra were acquired on Duetta Fluorescence and Absorbance Spectrometer (Horiba, Inc., Chicago, IL, USA). The Duetta device was also used for absorbance measurements. Fluorescence measurements were carried out in Sub-Micro Fluorometer Cell, model 16.40F-Q-10 (from StarnaCells, Inc., Atascadero, CA, USA) at a room temperature of ~22 °C. The excitation–emission matrix spectra (EEMS) were recorded at a resolution of 0.5 nm, and fluorescence spectra were recorded with an emission wavelength range from 300 nm to 1000 nm. Matrix data were then used for a 2D contour plot using the MagicPlot Pro software (St. Petersburg, Russia). The expected fluorescence, F_0_, was then adjusted using an appropriate dilution factor. The electrochemical response of the bare SPE and modified SPE electrodes was performed by CV and potentiometric techniques. The three-electrode cell was used for CV measurements, while a two-electrode cell was used for potentiometric measurements at an open circuit. A PBS buffer solution, at a pH of 7.4, was used as the electrolyte solution. After the measurements’ stabilization, a glucose solution was added in successive amounts into the electrochemical cell. The biocathode-generated signal was expressed as the change in cathodic current or the change in potential calculated from the signal recorded by the addition of the glucose solution. During the CV measurements, the potential was scanned in the range of −2 to +1.8 V at a scan rate of 100 mV/s, and the peak current was monitored using a potentiostat (Ossila, T2006B1-US).

### 2.3. Synthesis of Ag-DNA Nanoclusters

The preparation of DNA-templated AgNCs, using a hairpin DNA structure with 12-cytosine loop, was carried out by mixing the template and aqueous AgNO_3_ and incubating the mixture for 25 min at room temperature in an ammonium acetate buffer (100 mM NH_4_OAc, pH 6.9). Further, a NaBH_4_ aqueous solution was added, and the samples were placed on ice and stirred vigorously. The AgNCs were allowed to mature in the dark at 4 °C for 12 h. The detailed step-by-step synthesis procedure can be found in our previous publications [27,29].

### 2.4. SPE Electrode Pretreatment and Preparation of the Biocathode

Screen-printed (SP) DRP-110 electrodes (SP), were purchased from Metrohm DropSens (Oviedo, Spain). The working surface area of the SP working electrode was A = 0.12566 cm^2^. The SP electrodes were used to perform the electrochemical synthesis of the AgNCs–PPCA/PPCA composite using cyclic voltammetry. Under the optimized conditions, the SP electrode and reference and auxiliary electrodes were placed in a 3 mL solution of AgNCs@DNA and 35.0 mM PCA in an electrochemical cell. The potential was then scanned at a rate of 100 mV/s for 50 cycles in a range of potentials from −2.0 V to +1.8 V. Thus, a composite of AgNCs embedded inside a PPCA shell was made on the surface of the SP working electrode. After synthesizing the composite electrode (AgNCs–PPCA), the electrode was washed with water. The electrode was next placed in the electrochemical cell filled with 3 mL of a solution containing the PBS buffer solution and 200.0 mM PCA for synthesizing a thin layer of PPCA. The potential was then scanned at a rate of 100 mV/s for 5 cycles in a range of potentials from −2.0 V to +1.8 V. To modify the glucose oxidase enzyme (GOx) over the modified SP working electrode, it was immersed in a mixture of 0.2 M EDC and 0.1 M NHS solutions and left for 30 min at ambient temperature. The electrode was washed with DI water and immersed immediately in a 10 mg/mL GOx solution. The GOx was attached covalently by an amide bond to the top surface layer of the SP electrode (AgNCs–PPCA/PPCA–GOx).

### 2.5. Optical Measurements

For optical detection, glucose was mixed with 100 μL of AgNCs@DNA, 5 μL of GOx (1 mM) via successive additions of 5 μL glucose (500 µM solution). The excitation–emission matrix spectra (EEMS) were recorded at a resolution of 0.5 nm. Fluorescence spectra were recorded with an emission wavelength range from 300 nm to 1000 nm. Matrix data were then used for a 2D contour plot using the MagicPlot Pro software (St. Petersburg, Russia). Fluorescence quenching measurements were performed with freshly prepared samples by adding small volumes of the glucose solution. The expected fluorescence, F_0_, was then adjusted using an appropriate dilution factor. For the reversible recovery of oxidized dark AgNCs, a NaBH_4_ aqueous solution was added in excess until maximum fluorescence signal recovery was achieved.

### 2.6. Electrochemical Measurements

For the electrochemical detection of glucose, the SP working electrode modified with AgNCs was first prepared using a three-electrode CV system. The stepwise modification process was characterized by cyclic voltammetry (CV) in the PBS and PCA solutions, respectively. Furthermore, the modified SP working electrode (AgNCs–PPCA/PCA) was used for the deposition of 10 μL of GOx and titrated with a glucose solution of a concentration of 10 mM in 5 μL increments. The CV signals were measured in a PBS solution for output signals. A two-electrode electrochemical cell consisting of Ag/AgCl was used as a reference electrode and AgNCs–PPCA/PCA–GOx was used as a working electrode for evaluating cell performance at the open-circuit potential.

## 3. Results

### 3.1. Detection of Glucose Using Optical Response of Redox-Active AgNCs@DNA (Ag_10_NC@hpC_12_)

DNA-templated silver nanoclusters (AgNCs@DNA) possess unique optical properties compared to regular organic fluorophores or quantum dots (QDs) [46]. AgNCs@DNA have tunable fluorescence, which depends on the templating DNA sequence. Additionally, their biocompatibility and nontoxic nature make AgNCs suitable for various applications including biosensing and bioimaging [29,47].

More importantly, AgNCs possess redox activity upon their interaction with oxidative species such as oxygen or hydrogen peroxide (H_2_O_2_) due to the partial charge of the AgNC [29,47]. Herein, we have used Ag_10_NC@hpC_12_, a previously reported nanocluster consisting of 10 silver atoms within a DNA hairpin loop structure in which the loop consists of 12 cytosines [48]. This nanocluster has an elongated shape and possesses a positive charge due to the partial reduction of DNA-protected silver atoms [48]. The change in the charge state, Ag_N_^0^/Ag_(10-N)_^+^, of the AgNC due to oxidation also changes the pattern of fluorescence from red (visible) to near-IR (invisible), as schematically shown in Figure 1A–D. The electronic structure also changes due to the removal of two electrons from the HOMO during oxidation, which reshapes the configuration of the outermost frontier orbitals (as schematically shown in Figure 1E). We hypothesize that such a reshaping yields either a “dark” nonemissive oxidized state or “red” emissive reduced state [49].

We have used this property of AgNCs@hpC_12_ to construct a biosensor based on the change in fluorescence intensity as an optical readout for the detection of glucose. The biosensor involves a glucose oxidase (GOx) enzyme, which reacts with glucose, producing gluconic acid and H_2_O_2_ during the oxidation of glucose.
(1)$$Glucose+\frac{1}{2}O_{2}+H_{2}O=Gluconic\;acid+H_{2}O_{2}$$

Further, AgNC@hpC_12_ interacts with H_2_O_2_ undergoing the oxidation process that changes the ratio of Ag^0^ to Ag^+^ in the nanocluster, Ag_N_^0^/Ag_(10-N)_^+^. This leads to the disappearance of initially strong “red” fluorescence (Figure 1). The emission pattern of the as-synthesized AgNC@hpC_12_ nanocluster initially has two peaks. The most intense emission peak (“red”) for AgNC@hpC_12_ has a maximum of excitation (λ_EXC_) if 555 nm and a maximum of emission (λ_EM_) of 630 nm (Figure 2A). The second peak is of a small intensity with λ_EXC_ = 485 nm and λ_EM_ = 595 nm (“orange”, Figure 2A). Only the “red” peak is susceptible to oxidation, changing its intensity in the presence of oxidative species (Appendix A). The intensity of the “orange” peak stays almost constant no matter the conditions, suggesting different redox properties of the two populations of AgNCs. We used the optical response of the “red” fluorescing AgNCs as a readout for detection purposes. Figure 2B shows the spectral pattern after the oxidation of “R”-AgNC@hpC_12_ with an excess of free H_2_O_2_. The complete disappearance of the “red” emission indicates effective interaction between “R”-AgNC@hpC_12_ and H_2_O_2_. This observation suggests the fast response of emissive “R”-AgNC@hpC_12_ to oxidative species, thus justifying its use as a detection strategy. We have further established that the “red” emission is recoverable by reduction with NaBH_4_ (Figure 1A,B), suggesting that the “R” state of AgNC@hpC_12_ is not destroyed but converted to an invisible (“dark”) state. Based on our recent theoretical study of AgNC@hpC_12_ optical properties, we hypothesize that AgNC@hpC_12_ oxidation stimulates an “R” to “dark” switch of states resulting in the apparent quenching of red fluorescence [48]. The quenching rate depends strongly on the amount of oxidizing agent present, thus providing a nice optical response mechanism of detection. The reversible recovery of the “R” peak is beneficial since the constructed biosensor can be made to be reusable by simply reducing the oxidized “R”-AgNC@hpC_12_.

Combining a mixture of GOx and AgNC@hpC_12_ with various concentrations of glucose between 0–250 μM results in a gradual decrease in fluorescence intensity for the “R”-AgNC@hpC_12_
Figure 2C,D and Appendix A). While the intensity of the “orange” peak remains almost the same, the intensity of the “red” peak decreases with the increasing intensity of glucose. We inferred that such a change could be used to detect glucose levels in a sample. Figure 2E shows the Stern–Volmer plot in which the F_0_/F ratio is plotted versus the concentration of glucose. F is the intensity of fluorescence at a particular probed glucose concentration and F_0_ is the intensity of fluorescence with no glucose added. The plot has two apparent regimes: (1) an apparent linear regime in the range of glucose concentrations of between 0 and ~150 μM and (2) a regime in which, above ~150 μM, the fluorescence ratio, F_0_/F, gradually increases. Hydrogen peroxide is a well-known collisional quencher, and a linear Stern–Volmer plot should in principle be expected [50]. While it is possible to fit a straight line to the data points in the range between 0–150 μM producing the Stern–Volmer constant, K_SV_ = 7442 M^−1^, it is obvious that the quenching has a more complicated character. The dependence of F_0_/F on C_glucose_ has an apparent upward tilt, indicating that this dependence cannot be explained by classical collisional quenching. The non-linearity of the plot and its upward tilt suggests that other mechanisms of quenching might be also present [50]. The range between 0 and 250 μM can be fitted with Equation (2):(2)$$\frac{F_{0}}{F}={\left(1+K_{Q}\times \left[C_{glucose}\right]\right)}^{2}$$

Typically, an upward tilt of the Stern–Volmer curve is interpreted as simultaneous competitive quenching due to multiple mechanisms, for example dynamic and static quenching [51,52]. Since it is not possible to distinguish between multiple quenching pathways from the data presented, the modified Stern–Volmer equation, Equation (2), was used to approximate the data points. Equation (2) fits the data points very well in the C_glucose_ range between 0 and 250 μM with the constant K_Q_ having the value of ~3495 M^−1^ as determined from the fitting of the data points. This observation may indicate that a complex quenching mechanism is afforded by hydrogen peroxide, which is a product of enzymatic reaction between glucose and Gox [27,50].

While more details will be required to fully describe the quenching mechanism, it is still possible to use AgNCs@DNA and their emission quenching as an optical readout for glucose detection in the low range of glucose concentrations, 0–250 μM. The relative change in intensity for the “red” peak is proportional to the amount of oxidative species, H_2_O_2_, generated in the reaction, which is directly proportional to the amount of glucose in the sample. Using the data of the plot, we determined the limit of detection (LOD) of the method as being three standard deviations above the background intensity which resulted in an LOD concentration of glucose of 22.8 μM.

### 3.2. Detection of Glucose Using Electrochemical Readout of AgNC Redox Response

AgNCs appear to be capable of donating and accepting electrons easily during the oxidation and reduction processes which are also associated with the color change of the Ag_10_NC@hpC_12_ emissive states. Additionally, the redox process of Ag_10_NC@hpC_12_ is also reversible, as we demonstrated using H_2_O_2_ and NaBH_4_ (Figure 1). Due to the redox activity of silver nanoclusters, they can be used as probes for the electrochemical mechanism of detection.
(3)$${\left(Ag_{10}NC\right)}^{N+}+H_{2}O_{2}={\left(Ag_{10}NC\right)}^{\left(N+2\right)+}+2OH^{-}$$

Equation (3) shows the overall balancing of electron transfer in the reaction of H_2_O_2_ with AgNCs. The balance indicates that one molecule of H_2_O_2_ draws two electrons from the Ag_10_NC@hpC_12_, increasing the total positive charge of the nanocluster by two. Since there is a definite number of charges transferred per molecule of H_2_O_2_, it is therefore possible to track the amount of glucose, since one molecule of glucose produces one molecule of H_2_O_2_ (see Equation (1)).

We have created an electrochemical biosensing system based on a novel biocathode strategy introduced recently [53]. Our design is an improved adaptation of this original strategy which involves the modification of a working electrode in the three-electrode setup of the screen-printed electrode with AgNCs@DNA, poly(pyrrole-2-carboxylic acid)-PPCA, and glucose oxidase (GOx). Figure 3A shows an illustration of the design. Unlike the common two-enzyme systems, this strategy only modifies a carbon cathode to produce a single-enzyme biofuel cell powered by glucose [53]. The cell also has electrochemical detection capability due to the use of AgNCs@DNA which become oxidized by H_2_O_2_ (a product of the GOx + glucose reaction; Equation (1)) and become reduced electrochemically during the cyclic voltammogram (CV) cycle. Additionally, the use of AgNCs@DNA replaces the requirement for the second enzyme, peroxidase (PO), typically attached to a cathode [54]. A single enzyme required for the operation of the biosensor, GOx, is covalently attached to the top of the PPCA polymer layer, further removing the need for a membrane of the cell. AgNCs@DNA are embedded within the PPCA layer of the cathode to react promptly with the H_2_O_2_ produced when GOx quickly processes glucose molecules at the outer surface of the electrode. Polymer-embedded AgNCs serve as charge transport mediators effectively shuttling the charges through the polymer from the outer layer modified with GOx to the surface of the electrode. All modifications are made on a miniature screen-printed electrode with the three-electrode system: a carbon counter electrode (CCE), carbon working electrode (CWE), and silver reference electrode (SRE).

The design responds very well to glucose as a fuel-generating electrical signal. We used cyclic voltammetry (CV) to monitor the biosensor’s response to the presence of glucose in a sample. Figure 3B shows a set of CV curves measured at concentrations of glucose varying from 0–0.5 mM. The CV curves feature three apparent peaks: (1) an anodic peak at E_a_ = 0.11 V, (2) a cathodic peak at E_1c_ = 0.73 V, and (3) a cathodic peak at E_2c_ = −0.75 V. The cathodic peak at E_1c_ = 0.73 V is also present in the control measurements with PBS only. Additionally, the anodic peak is more pronounced and has sharper features; therefore, we further used E_a_ and I_pa_ to evaluate the biosensor’s performance. Figure 3B shows that the increase in glucose concentration raises the level of I_pa_ as well as slightly shifting E to the higher values of potential. The values of the peak current, I_pa_, were then used to calculate the current density, which takes into account the area of the working electrode (A = 0.12566 cm^2^), as shown in Figure 3C. Further, the power density was also calculated from the available data, as shown in Figure 3D.

The current density has a nonlinear dependence in response to the glucose concentration in the sample in the range between 0 and 6 mM of glucose. The growth of the current slows down and gradually reaches a plateau at higher concentrations of glucose. The behavior of I_pa_ dependence on glucose concentration suggests that the electrochemical processes are controlled by the enzymatic reaction, in which GOx reacts with glucose. In general, the Michaels–Menten kinetics model should be considered for these types of reactions, and a clear separation of analytical regimes for analytes or enzymes should be made. In the case of GOx and glucose, we defined the C_glucose_ range spanning0 to 1 mM (the highlighted area in Figure 3C,D) as the analytical range suitable for the characterization of glucose. Therefore, this range was used to determine the detection limit (LOD) of the electrochemical readout for glucose which was found to be 29.2 μM (see Appendix A for details).

## 4. Discussion

Using nanomaterials for the construction of biosensors provides a range of advantages for the next generation of sensing devices [1]. These advantages lead to the better sensitivity and specificity of novel biosensor designs. Nanomaterials can also enable miniaturization and provide biosensors with multifunctional capabilities. DNA-templated silver nanoclusters are advanced nanomaterials consisting of only a few silver atoms wrapped around and stabilized by the cytosine-rich DNA sequence. AgNCs@DNA exhibit unique properties mostly associated with their electronic-state structure. The increased density of electronic states due to the quantum confinement at this size results in discrete energy levels. These electronic properties then largely define both the optical properties and redox properties of AgNCs@DNA. The properties can be tuned by controlling the nanoclusters’ size, shape, and charge state [29,47]. However, the ultimate control of these properties requires an incredible level of control over the exact positioning of Ag atoms at an individual level [48]. AgNCs@DNA can be synthesized with different DNA sequences resulting in different energy gaps between HOMO–LUMO. Learning how to gain control over the AgNC structure will enable the regulation of optical and electrochemical properties. The controllable design of various types of biosensors should thus be expected. Since the properties of AgNCs can be tuned, this can possibly lead to multiple colors and various redox properties which can be tailored to a specific application or a specific biosensor design. Therefore, many benefits and future potentials may be offered due to the multifunctionality provided by combining two readout strategies.

AgNCs@hpC_12_ only contains 10 atoms of silver, adapting a “big-dipper” conformation within the C_12_ loop of the DNA hairpin with explicit frontier orbital transitions (HOMO->LUMO, HOMO-1->LUMO, and HOMO->LUMO+1) [48]. The energy of these transitions not only defines the excitation and emission wavelengths but also the redox properties of AgNCs@DNA [48]. This type of advanced nanomaterial is, therefore, suitable for the construction of novel biosensors which are capable of utilizing multiple readout strategies simultaneously. Due to the partial charge established on AgNCs@hpC_12_, two readout strategies are possible—optical (fluorescence) and electrochemical (current). The fluorescence readout relies on the promptness with which photons are emitted by AgNCs@DNA. As such, this readout requires an optical setup that is ready to detect the photons emitted. The second strategy relies on the role AgNCs@DNA play in electrochemical detection. AgNCs@DNA mediate the charge transfer between the redox enzyme and the electrode, which is essential to achieve the desired sensing performance.

For the study presented here, charge mediation was established between glucose oxidase, GOx, and a carbon electrode in a single-enzyme BFC design, as shown in Figure 3A. Generally, glucose BFCs utilize two enzyme-based systems [55], which makes the system more complex and cost-effective. Additionally, the enzymes used may have different optimal operating conditions. These drawbacks can be avoided by designing a single-enzyme BFC, which uses the same enzyme for the anodic and cathodic reactions [56]. Recently, a single-enzyme design has been introduced which only requires the modification of a cathode with the involvement of an effective redox entity as a signal transducer [53]. We have adopted the strategy to include the modification of the biocathode with AgNCs@DNA as the redox entity, creating a single-enzyme self-powered biosensor. Incorporation of AgNCs@DNA at the electrode in a thin layer of a PPCA polymer protects the nanoclusters from oxidation by ambient oxygen while also allowing reversible reduction and oxidation. The operation of the biocathode provides electrochemical detection capability due to the use of AgNCs@DNA. AgNCs@DNA embedded in the PPCA layer of the cathode rapidly react with H_2_O_2_ generated by GOx in the presence of glucose. AgNCs@DNA act as charge transport mediators, transporting charges from the outer modified polymer layer with GOx to the surface of the electrode [57]. During this process, biorecognition events are converted into a measurable signal—current. Therefore, the design presented here is based on a single-enzyme BFC in which glucose, in addition to being an analyte, also serves as a fuel for the cell’s performance.

In addition to providing an advanced operation mode BFC, the design also offers a low limit of glucose detection. The limits of detection (LOD) demonstrated in this study extend to very low levels of glucose, ~23 μM and ~29 μM, for optical and electrochemical readout strategies, respectively. These concentrations are much lower than the levels of glucose expected in any biological fluid, including blood (3.9 to 5.6 mM), urine (<800 μM) [58], sweat (<200 μM) [59], and tears (<200 μM) [60]. It is quite impressive that the LODs are orders of magnitude lower than the biologically relevant concentrations, demonstrating the superb capabilities of the proposed sensing mechanism. What is also important is that the design presented here functions as a biofuel cell (BFC) powered by glucose, which is an analyte also. The chemical energy of glucose is converted to electrical energy via several components of the biosensor design coupled together. The power density (Figure 3D) dependence on glucose concentration, together with the open circuit potential (OCP) plot versus glucose concentration (Appendix A), suggest the effective operation of the design as a biofuel cell. Both the potential and power show similar dependence, gradually increasing with an increased concentration of glucose. Although the analytical range suitable for the detection of glucose is limited to 0–250 μM, our design of a biofuel cell keeps producing power at higher concentrations of glucose. BFC-based operation has the prospects of eliminating the need for external power, bringing biosensors a step closer to the construction of self-sufficient and autonomous devices. This can be particularly useful for remote operation and low-resource environments.

## 5. Conclusions

In summary, we have demonstrated the use of AgNCs@DNA in the construction of a novel dual-mode biosensor. Utilizing unique properties of the nanoclusters, two readout strategies, optical and electrochemical, can be utilized. We have characterized the performance of the biosensor, demonstrating very small limits of detection: as low as 23 μM and 29 μM of glucose can be detected with optical and electrochemical strategies, respectively. Furthermore, this novel design of the biosensor employs a single-enzyme membraneless biofuel cell. The BFC is powered by glucose, the analyte, thus requiring no external power. While we demonstrated the applicability of the design in detecting glucose, other analytes could also be detected. If GOx is replaced or combined with another bioenzyme, other biofuel sources could be utilized. This suggests the possibility of the broader impact of the developed strategy on the efficient utilization of biofuel cell design for powering sensitive and specific detection, personalized medicine, and remote autonomous biosensor operation.

## Figures and Tables

**Figure 1 nanomaterials-13-01299-f001:**
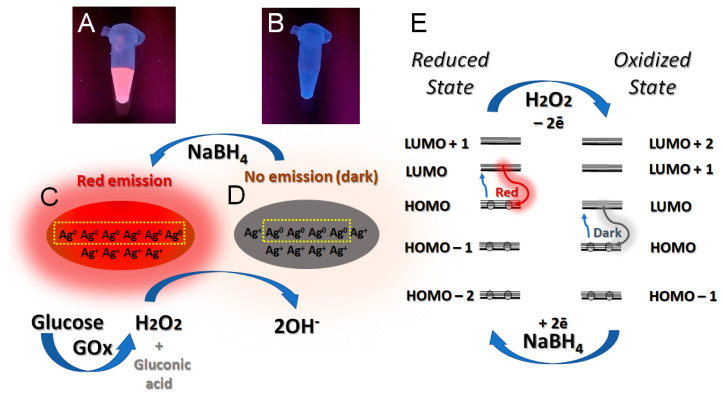
(**A**) Fluorescence of AgNC@hpC_12_ sample excited by UV on a transilluminator, (**B**) fluorescence of AgNC@hpC_12_ sample excited by UV on a transilluminator after addition of excess H_2_O_2_, (**C**) schematic representation of the nanocluster’s structure and composition before oxidation, (**D**) schematic representation of the nanocluster’s structure and composition after oxidation, (**E**) schematic representation of the effect of oxidation–reduction on the redistribution of electrons showing new possibility for excitation and emission of “dark” AgNCs.

**Figure 2 nanomaterials-13-01299-f002:**
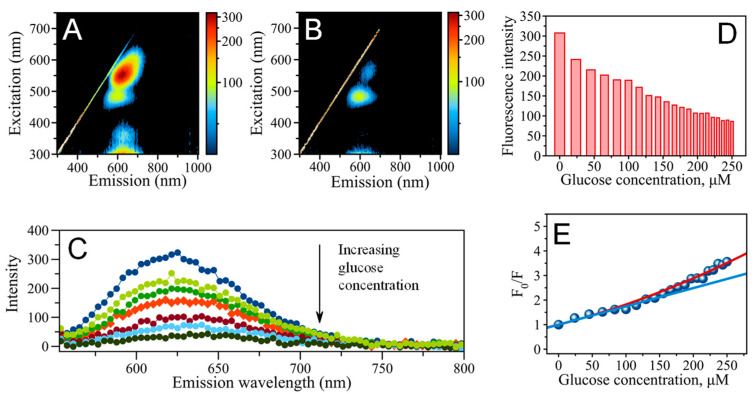
(**A**) EEMS showing emission pattern of AgNC@hpC_12_ mixed with GOx but without added glucose, (**B**) emission–excitation map showing emission pattern of AgNC@hpC_12_ mixed with GOx with ~230 μM glucose added, (**C**) fluorescence spectra of AgNC@hpC_12_ (λ_EXC_ = 550 nm) with successive additions of glucose, the spectra show a drop in intensity of fluorescence upon increase of C_glucose_ in the range between 0–250 μM, (**D**) fluorescence intensities of AgNC@hpC_12_ in response to glucose addition, and (**E**) Stern–Volmer, F_0_/F vs. C_glucose_, plot; solid blue line is a linear fit in the C_glucose_ range of 0–150 μM; solid red line is a fit with Equation (2).

**Figure 3 nanomaterials-13-01299-f003:**
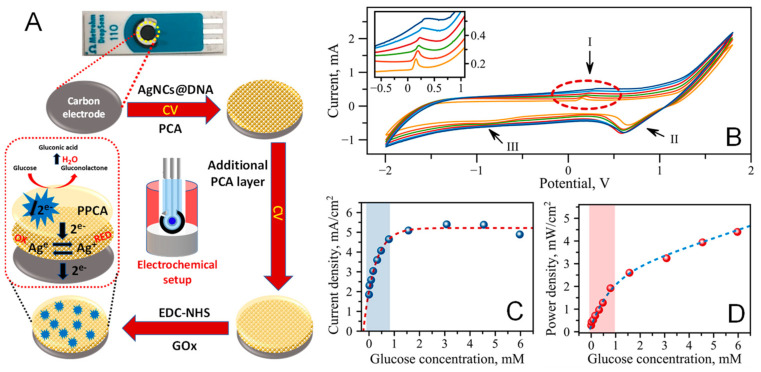
(**A**) Design of the single-enzyme, AgNCs-based three-electrode biosensing system, and (**B**) cyclic voltammograms of the modified electrodes in the solution with varied concentrations of glucose ranging from 0 to 0.5 mM ( orange: 0 mM; red: 0.1 mM; green: 0.2 mM; ruby: 0.3 mM; light blue: 0.4 mM; and blue: 0.5 mM); arrows indicate the positions of three prominent peaks: (I) E_a_ = 0.11 V, (II) E_1c_ = 0.73 V, and (III) E_2c_ = −0.75 V. Parameters: PBS, v = 100 mV/s; E_range_ = from −2.0 to 1.8 V; Inset: enlarged portion of CV curves at E_a_ = 0.11 V; (**C**) current density dependence on glucose concentration (dashed nonlinear regression curves are plotted as a guide to the eye); (**D**) power density, P in mW/cm^2^, dependence on glucose concentration.

## Data Availability

The data reported in this article are available upon request.

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
