# Peer review of "DNA-Templated Silver Nanoclusters as Dual-Mode Sensitive Probes for Self-Powered Biosensor Fueled by Glucose"

_nanomaterials, 2023, doi:10.3390/nano13081299_

Round 1

Reviewer 1 Report

The article is devoted to the design and construction of new glucose sensors based on the reaction of the sensor material (silver nanoclusters on DNA) to the level of hydrogen peroxide produced by the glucose oxidation by glucose oxidase GOx. The article uses two approaches for sensing - optical and electrochemical. Both methods showed a comparable detection limit. The article is written in good language and sounds scientifically.

The authors justify their work well. Of particular interest is the electrochemical approach to detect glucose, since it is possible to create an energy-independent biofuel cell based on the presented electrode, the power density of which will make it possible to carry out online measurements of glucose in vivo. Although the fluorescent response of the sensor to glucose is limited to 0-250 µM, which is insufficient for in vivo measurements, this will not interfere with in vitro glucose measurements.

The article may arouse keen interest among scientists and specialists in various fields of science.

Comments of reviewer:

Equation 2. The applicability of this equation to explain the mixed quenching mechanism is questionable. The dynamic constant is valid when a linear decrease in the lifetime of the excited state is observed, and it cannot be determined in the experiment described by the authors. If it was an approximation of experimental data, it should be explained in the article or in the supporting info in any case. Thus, the constant KD cannot be called dynamic.

Equation 3. Let’s consider N = 8. So, according to the Equ. 3 the balance of charges will be as follows:

8+ + 0 = (8-2)+ + 2-. Finally, 8+ = 4+. Equation needs to be corrected. Positively multi charged particles cannot donate electrons to the oxidizer.

Reviewer questions:

Could the authors provide more information about the limits of applicability of this Equation 2 and where it comes from. The equality of dynamic and static constants also looks strange. How were they measured?

Figure 1E. Will the -2e- transition in the molecule cause the transformation HOMO HOMO -2, and not HOMO -1, as on the Figure 1? What will happen to the energy gap in this case, and is it possible to confirm this with at least some quantum calculations?

Conclusion:

I recommend publishing the article after minor revision.

Author Response

We greatly appreciate the reviewer’s comments specifically on both equations which allowed us to revisit them and fix misprints. Below are our point-by-point responses highlighted in italics.

Reviewer #1:

The article is devoted to the design and construction of new glucose sensors based on the reaction of the sensor material (silver nanoclusters on DNA) to the level of hydrogen peroxide produced by the glucose oxidation by glucose oxidase GOx. The article uses two approaches for sensing - optical and electrochemical. Both methods showed a comparable detection limit. The article is written in good language and sounds scientifically.

The authors justify their work well. Of particular interest is the electrochemical approach to detect glucose, since it is possible to create an energy-independent biofuel cell based on the presented electrode, the power density of which will make it possible to carry out online measurements of glucose in vivo. Although the fluorescent response of the sensor to glucose is limited to 0-250 µM, which is insufficient for in vivo measurements, this will not interfere with in vitro glucose measurements.

The article may arouse keen interest among scientists and specialists in various fields of science.

Comments of reviewer:

Equation 2. The applicability of this equation to explain the mixed quenching mechanism is questionable. The dynamic constant is valid when a linear decrease in the lifetime of the excited state is observed, and it cannot be determined in the experiment described by the authors. If it was an approximation of experimental data, it should be explained in the article or in the supporting info in any case. Thus, the constant KD cannot be called dynamic.

We have removed KD from the description as suggested by the reviewer. Indeed, it is not possible to determine KD from the data presented. We have modified the equation to leave the second-order dependence on glucose concentration since it approximates the measured data points very well and combined two constants into one KQ. Therefore, we have corrected Equation #2 in the revision to remove KD, and reiterated that the modified Stern-Volmer equation, Eq. 2, was used only to approximate the data points.

Equation 3. Let’s consider N = 8. So, according to the Equ. 3 the balance of charges will be as follows:

8+ + 0 = (8-2)+ + 2-. Finally, 8+ = 4+. Equation needs to be corrected. Positively multi charged particles cannot donate electrons to the oxidizer.

We would like to thank the reviewer for pointing out this apparent misprint, the equation should have (N+2)+ superscript on the right side of the equation rather than (N-2)+. Now everything balances out: 8++0 = (8+2)+ + 2- , so 8+ =8+. We have corrected Equation #3 in the revised manuscript.

Reviewer questions:

Could the authors provide more information about the limits of applicability of this Equation 2 and where it comes from. The equality of dynamic and static constants also looks strange. How were they measured?

We believe that the limits of applicability for Equation #2 come from our incomplete understanding of the quenching mechanisms. As it is apparent from the Stern-Volmer plot - multiple quenching mechanisms are present. We have used the modified Stern-Volmer equation to fit the data points, which did provide a good approximation within 0-250 μM range of glucose concentration. However, as the reviewer correctly pointed out, time-resolved fluorescence may be needed to distinguish dynamic and static mechanisms further. Additionally, multiple “dark” states may be present, providing multiple channels for quenching with their own quenching constants. We, therefore, refrained from making unsupported claims and stayed within the applicability of the two readouts, optical and electrochemical, for detection purposes. We hope that further studies, including computational, will reveal more details of the quenching mechanism allowing us to have a more precise description of the data dependence.  

Figure 1E. Will the -2e- transition in the molecule cause the transformation HOMO → HOMO -2, and not HOMO -1, as on Figure 1? What will happen to the energy gap in this case, and is it possible to confirm this with at least some quantum calculations?

The diagram in Figure 1 refers to the relative position of HOMO-n and LUMO+n orbitals relative to HOMO and LUMO, respectively. For example, HOMO-1 refers to the orbital right below HOMO. Since each orbital hosts two electrons, the loss of 2e- from HOMO (left side of the graph) will result in HOMO -> HOMO-1 transition (right side of the graph). The reviewer is absolutely correct that the energy gap will change as a result of 2e loss. We have just begun an extensive computational characterization of AgNCs’ structures and associated energy transitions and hope to be able to uncover more details of these interesting systems in our future publications.  

 Conclusion:

I recommend publishing the article after minor revision.

Reviewer 2 Report

Glucose, which plays an important role in biofuel cells, was detected using DNA-templated silver nanoclusters (AgNCs@DNA). The AgNC@DNA emits fluorescence through a redox reaction with glucose. The redox reaction can be quantified with current density, and the fluorescence and current density are dependent on glucose concentration. The limit of detection (LOD) of glucose estimated by fluorescence is similar to that by current density. The dual-mode analysis can be utilized for detecting dilute glucose as compared to glucose in body fluids. Thus, I can recommend publishing this manuscript after addressing minor comments as follows.

Fig. 3B: It is unclear which colored curves correspond to the CV curves at each concentration of glucose. It is suggested to show the peaks at 0.11, 0.73, and -0.75 V in the figure.

Fig. 3C/D: An enlarged graph of low concentrations (sub-mM) should be displayed to estimate the LOD.

Author Response

We would like to thank the reviewer for the comments, we believe we have fully addressed all the concerns, which in turn improved our paper as well. Below are our point-by-point responses highlighted in italics.

Reviewer #2:

Glucose, which plays an important role in biofuel cells, was detected using DNA-templated silver nanoclusters (AgNCs@DNA). The AgNC@DNA emits fluorescence through a redox reaction with glucose. The redox reaction can be quantified with current density, and the fluorescence and current density are dependent on glucose concentration. The limit of detection (LOD) of glucose estimated by fluorescence is similar to that by current density. The dual-mode analysis can be utilized for detecting dilute glucose as compared to glucose in body fluids. Thus, I can recommend publishing this manuscript after addressing minor comments as follows.

Fig. 3B: It is unclear which colored curves correspond to the CV curves at each concentration of glucose. It is suggested to show the peaks at 0.11, 0.73, and -0.75 V in the figure.

The color code is now described in the figure captions and the peaks are indicated with arrows.

Fig. 3C/D: An enlarged graph of low concentrations (sub-mM) should be displayed to estimate the LOD

An enlarged graph of Power Density vs glucose concentration at a low concentration range, 0-1mM, is now included in supplementary materials as Figure S3 with the linear fit and details.